# Variational Bayes on Monte Carlo Steroids

**Aditya Grover, Stefano Ermon**
Department of Computer Science
Stanford University
{adityag,ermon}@cs.stanford.edu

## Abstract

Variational approaches are often used to approximate intractable posteriors or normalization constants in hierarchical latent variable models. While often effective in practice, it is known that the approximation error can be arbitrarily large. We propose a new class of bounds on the marginal log-likelihood of directed latent variable models. Our approach relies on random projections to simplify the posterior. In contrast to standard variational methods, our bounds are guaranteed to be tight with high probability. We provide a new approach for learning latent variable models based on optimizing our new bounds on the log-likelihood. We demonstrate empirical improvements on benchmark datasets in vision and language for sigmoid belief networks, where a neural network is used to approximate the posterior.

## 1 Introduction

Hierarchical models with multiple layers of latent variables are emerging as a powerful class of generative models of data in a range of domains, ranging from images to text [1, 18]. The great expressive power of these models, however, comes at a significant computational cost. Inference and learning are typically very difficult, often involving intractable posteriors or normalization constants.

The key challenge in learning latent variable models is to evaluate the marginal log-likelihood of the data and optimize it over the parameters. The marginal log-likelihood is generally non-convex and intractable to compute, as it requires marginalizing over the unobserved variables. Existing approaches rely on Monte Carlo [12] or variational methods [2] to approximate this integral. Variational approximations are particularly suitable for *directed* models, because they directly provide tractable *lower bounds* on the marginal log-likelihood.

Variational Bayes approaches use variational lower bounds as a tractable proxy for the true marginal log-likelihood. While optimizing a lower bound is a reasonable strategy, the true marginal log-likelihood of the data is not necessarily guaranteed to improve. In fact, it is well known that variational bounds can be arbitrarily loose. Intuitively, difficulties arise when the approximating family of tractable distributions is too simple and cannot capture the complexity of the (intractable) posterior, no matter how well the variational parameters are chosen.

In this paper, we propose a new class of marginal log-likelihood approximations for directed latent variable models with discrete latent units that are guaranteed to be tight, assuming an optimal choice for the variational parameters. Our approach uses a recently introduced class of *random projections* [7, 15] to improve the approximation achieved by a standard variational approximation such as mean-field. Intuitively, our approach relies on a sequence of *random projections* to simplify the posterior, without losing too much information at each step, until it becomes easy to approximate with a mean-field distribution.

We provide a novel learning framework for directed, discrete latent variable models based on optimizing this new lower bound. Our approach jointly optimizes the parameters of the generative model and the variational parameters of the approximating model using stochastic gradient descent

(SGD). We demonstrate an application of this approach to sigmoid belief networks, where neural networks are used to specify both the generative model and the family of approximating distributions. We use a new stochastic, sampling based approximation of the variational projected bound, and show empirically that by employing random projections we are able to significantly improve the marginal log-likelihood estimates.

Overall, our paper makes the following contributions:

1. We extend [15], deriving new (tight) stochastic bounds for the marginal log-likelihood of directed, discrete latent variable models.
2. We develop a "black-box" [23] random-projection based algorithm for learning and inference that is applicable beyond the exponential family and does not require deriving potentially complex updates or gradients by hand.
3. We demonstrate the superior performance of our algorithm on sigmoid belief networks with discrete latent variables in which a highly expressive neural network approximates the posterior and optimization is done using an SGD variant [16].

## 2 Background setup

Let $p_\theta(\mathbf{X}, \mathbf{Z})$ denote the joint probability distribution of a directed latent variable model parameterized by $\theta$. Here, $\mathbf{X} = \{X_i\}_{i=1}^m$ represents the observed random variables which are explained through a set of latent variables $\mathbf{Z} = \{Z_i\}_{i=1}^n$. In general, $\mathbf{X}$ and $\mathbf{Z}$ can be discrete or continuous. Our learning framework assumes discrete latent variables $\mathbf{Z}$ whereas $\mathbf{X}$ can be discrete or continuous.

Learning latent variable models based on the maximum likelihood principle involves an intractable marginalization over the latent variables. There are two complementary approaches to learning latent variable models based on approximate inference which we discuss next.

### 2.1 Learning based on amortized variational inference

In variational inference, given a data point $\mathbf{x}$, we introduce a distribution $q_\phi(\mathbf{z})$ parametrized by a set of variational parameters $\phi$. Using Jensen's inequality, we can lower bound the marginal log-likelihood of $\mathbf{x}$ as an expectation with respect to $q$.

$$
\begin{aligned}
\log p_\theta(\mathbf{x}) = \log \sum_{\mathbf{z}} p_\theta(\mathbf{x}, \mathbf{z}) &= \log \sum_{\mathbf{z}} q_\phi(\mathbf{z}) \cdot \frac{p_\theta(\mathbf{x}, \mathbf{z})}{q_\phi(\mathbf{z})} \\
&\geq \sum_{\mathbf{z}} q_\phi(\mathbf{z}) \cdot \log \frac{p_\theta(\mathbf{x}, \mathbf{z})}{q_\phi(\mathbf{z})} = \mathbb{E}_q[\log p_\theta(\mathbf{x}, \mathbf{z}) - \log q_\phi(\mathbf{z})].
\end{aligned} \tag{1}
$$

The evidence lower bound (ELBO) above is tight when $q_\phi(\mathbf{z}) = p_\theta(\mathbf{z}|\mathbf{x})$. Therefore, variational inference can be seen as a problem of computing the parameters $\phi$ from an approximating family of distributions $Q$ such that the ELBO can be evaluated efficiently and the approximate posterior over the latent variables is close to the true posterior.

In the setting we consider, we only have access to samples $\mathbf{x} \sim p_\theta(\mathbf{x})$ from the underlying distribution. Further, we can *amortize* the cost of inference by learning a single data-dependent variational posterior $q_\phi(\mathbf{z}|\mathbf{x})$ [9]. This increases the generalization strength of our approximate posterior and speeds up inference at test time. Hence, learning using amortized variational inference optimizes the average ELBO (across all $\mathbf{x}$) jointly over the model parameters ($\theta$) as well as the variational parameters ($\phi$).

### 2.2 Learning based on importance sampling

A tighter lower bound of the log-likelihood can be obtained using importance sampling (IS) [4]. From this perspective, we view $q_\phi(\mathbf{z}|\mathbf{x})$ as a proposal distribution and optimize the following lower bound:

$$
\log p_\theta(\mathbf{x}) \geq \mathbb{E}_q \left[ \log \frac{1}{S} \sum_{i=1}^S \frac{p_\theta(\mathbf{x}, \mathbf{z}_i)}{q_\phi(\mathbf{z_i}|\mathbf{x})} \right] \tag{2}
$$

where each of the S samples are drawn from $q_\phi(\mathbf{z}|\mathbf{x})$. The IS estimate reduces to the variational objective for $S = 1$ in Eq. (1). From Theorem 1 of [4], the IS estimate is also a lower bound to the true log-likelihood of a model and is asymptotically unbiased under mild conditions. Furthermore, increasing $S$ will never lead to a weaker lower bound.

# 3 Learning using random projections

Complex data distributions are well represented by generative models that are flexible and have many modes. Even though the posterior is generally much more peaked than the prior, learning a model with multiple modes can help represent arbitrary structure and supports multiple explanations for the observed data. This largely explains the empirical success of deep models for representational learning, where the number of modes grows nearly exponentially with the number of hidden layers [1, 22].

Sampling-based estimates for the marginal log-likelihood in Eq. (1) and Eq. (2) have high variance, because they might "miss" important modes of the distribution. Increasing $S$ helps but one might need an extremely large number of samples to cover the entire posterior if it is highly multi-modal.

## 3.1 Exponential sampling

Our key idea is to use random projections [7, 15, 28], a hash-based inference scheme that can efficiently sample an exponentially large number of latent variable configurations from the posterior. Intuitively, instead of sampling a single latent configuration each time, we sample (exponentially large) *buckets* of configurations defined implicitly as the solutions to randomly generated constraints.

Formally, let $\mathcal{P}$ be the set of all posterior distributions defined over $\mathbf{z} \in \{0,1\}^n$ conditioned on $\mathbf{x}$. [1] A random projection $R_{A,b}^k : \mathcal{P} \to \mathcal{P}$ is a family of operators specified by $A \in \{0,1\}^{k \times n}, b \in \{0,1\}^k$ for a $k \in \{0,1,\ldots,n\}$. Each operator maps the posterior distribution $p_\theta(\mathbf{z}|\mathbf{x})$ to another distribution $R_{A,b}^k[p_\theta(\mathbf{z}|\mathbf{x})]$ with probability mass proportional to $p_\theta(\mathbf{z}|\mathbf{x})$ and a support set restricted to $\{\mathbf{z} : A\mathbf{z} = b \mod 2\}$. When $A, b$ are chosen uniformly at random, this defines a family of pairwise independent hash functions $\mathcal{H} = \{h_{A,b}(\mathbf{z}) : \{0,1\}^n \to \{0,1\}^k\}$ where $h_{A,b}(\mathbf{z}) = A\mathbf{z} + b \mod 2$. See [7, 27] for details.

The constraints on the space of assignments of $\mathbf{z}$ can be viewed as parity (XOR) constraints. The random projection reduces the dimensionality of the problem in the sense that a subset of $k$ variables becomes a deterministic function of the remaining $n - k$. [2] By uniformly randomizing over the choice of the constraints, we can extend similar results from [28] to get the following expressions for the first and second order moments of the normalization constant of the projected posterior distribution.

**Lemma 3.1.** *Given* $A \in \{0,1\}^{k \times n} \overset{iid}{\sim}$ Bernoulli$(\frac{1}{2})$ *and* $b \in \{0,1\}^k \overset{iid}{\sim}$ Bernoulli$(\frac{1}{2})$ *for* $k \in \{0,1,\ldots,n\}$, *we have the following relationships:*

$$\mathbb{E}_{A,b}\left[\sum_{\mathbf{z}:A\mathbf{z}=b \mod 2} p_\theta(\mathbf{x},\mathbf{z})\right] = 2^{-k}p_\theta(\mathbf{x}) \tag{3}$$

$$Var\left(\sum_{\mathbf{z}:A\mathbf{z}=b \mod 2} p_\theta(\mathbf{x},\mathbf{z})\right) = 2^{-k}(1 - 2^{-k})\sum_{\mathbf{z}} p_\theta(\mathbf{x},\mathbf{z})^2 \tag{4}$$

Hence, a typical random projection of the posterior distribution partitions the support into $2^k$ subsets or *buckets*, each containing $2^{n-k}$ states. In contrast, typical Monte Carlo estimators for variational inference and importance sampling can be thought of as partitioning the state space into $2^n$ subsets, each containing a single state.

There are two obvious challenges with this random projection approach:

1. What is a good proposal distribution to select the appropriate constraint sets, i.e., *buckets*?

2. Once we select a bucket, how can we perform efficient inference over the (exponentially large number of) configurations within the *bucket*?

Surprisingly, using a uniform proposal for 1) and a simple mean-field inference strategy for 2), we will provide an estimator for the marginal log-likelihood that will guarantee tight bounds for the quality of our solution. Unlike the estimates produced by variational inference in Eq. (1) and importance sampling in Eq. (2) which are stochastic lower bounds for the true log-likelihood, our estimate will be a provably tight approximation for the marginal log-likelihood *with high probability* using a small number of samples, *assuming we can compute an optimal mean-field approximation*. Given that finding an optimal mean-field (fully factored) approximation is a non-convex optimization problem, our result does not violate known worst-case hardness results for probabilistic inference.

## 3.2 Tighter guarantees on the marginal log-likelihood

Intuitively, we want to project the posterior distribution in a "predictable" way such that key properties are preserved. Specifically, in order to apply the results in Lemma 3.1, we will use a uniform proposal for any given choice of constraints. Secondly, we will reason about the exponential configurations corresponding to any given choice of constraint set using variational inference with an approximating family of tractable distributions $\mathcal{Q}$. We follow the proof strategy of [15] and extend their work on bounding the partition function for inference in *undirected* graphical models to the learning setting for *directed* latent variable models. We assume the following:

**Assumption 3.1.** *The set $\mathcal{D}$ of degenerate distributions, i.e., distributions which assign all the probability mass to a single configuration, is contained in $\mathcal{Q}$: $\mathcal{D} \subset \mathcal{Q}$.*

This assumption is true for most commonly used approximating families of distributions such as mean-field $\mathcal{Q}_{MF} = \{q(\mathbf{z}) : q(\mathbf{z}) = q_1(z_1) \cdots q_\ell(x_\ell)\}$, structured mean-field [3], etc. We now define a projected variational inference problem as follows:

**Definition 3.1.** *Let $A_t^k \in \{0,1\}^{k \times n} \overset{iid}{\sim}$ Bernoulli$(\frac{1}{2})$ and $b_t^k \in \{0,1\}^k \overset{iid}{\sim}$ Bernoulli$(\frac{1}{2})$ for $k \in [0, 1, \cdots, n]$ and $t \in [1, 2, \cdots, T]$. Let $\mathcal{Q}$ be a family of distributions such that Assumption 3.1 holds. The optimal solutions for the projected variational inference problems, $\gamma_t^k$, are defined as follows:*

$$\log \gamma_t^k(\mathbf{x}) = \max_{q \in \mathcal{Q}} \sum_{\mathbf{z}: A_t^k \mathbf{z} = b_t^k \bmod 2} q_\phi(\mathbf{z}|\mathbf{x}) \big( \log p_\theta(\mathbf{x}, \mathbf{z}) - \log q_\phi(\mathbf{z}|\mathbf{x}) \big) \tag{5}$$

We now derive bounds on the marginal likelihood $p_\theta(\mathbf{x})$ using two estimators that aggregate solutions to the projected variational inference problems.

### 3.2.1 Bounds based on mean aggregation

Our first estimator is a weighted average of the projected variational inference problems.

**Definition 3.2.** *For any given $k$, the mean estimator over $T$ instances of the projected variational inference problems is defined as follows:*

$$\mathcal{L}_\mu^{k,T}(\mathbf{x}) = \frac{1}{T} \sum_{t=1}^T \gamma_t^k(\mathbf{x}) 2^k. \tag{6}$$

Note that the stochasticity in the mean estimator is due to the choice of our random matrices $A_t^k, b_t^k$ in Definition 5. Consequently, we obtain the following guarantees:

**Theorem 3.1.** *The mean estimator is a lower bound for $p_\theta(\mathbf{x})$ in expectation:*

$$\mathbb{E}\left[\mathcal{L}_\mu^{k,T}(\mathbf{x})\right] \leq p_\theta(\mathbf{x}).$$

*Moreover, there exists a $k^\star$ and a positive constant $\alpha$ such that for any $\Delta > 0$, if $T \geq \frac{1}{\alpha}\left(\log(2n/\Delta)\right)$ then with probability at least $(1 - 2\Delta)$,*

$$\mathcal{L}_\mu^{k^\star, T}(\mathbf{x}) \geq \frac{p_\theta(\mathbf{x})}{64(n+1)}.$$

*Proof sketch:* For the first part of the theorem, note that the solution of a projected variational problem for any choice of $A_t^k$ and $b_t^k$ with a fixed $k$ in Eq. (5) is a lower bound to the sum $\sum_{\mathbf{z}:A_t^k\mathbf{z}=b_t^k \mod 2} p_\theta(\mathbf{x},\mathbf{z})$ using Eq. (1). Now, we can use Eq. (3) in Lemma 3.1 to obtain the upper bound in expectation. The second part of the proof extends naturally from Theorem 3.2 which we state next. Please refer to the supplementary material for a detailed proof.

### 3.2.2 Bounds based on median aggregation

We can additionally aggregate the solutions to Eq. (5) using the median estimator. This gives us tighter guarantees, including a lower bound that does not require us to take an expectation.

**Definition 3.3.** *For any given $k$, the median estimator over $T$ instances of the projected variational inference problems is defined as follows:*

$$\mathcal{L}_{Md}^{k,T}(\mathbf{x}) = Median\left(\gamma_1^k(\mathbf{x}),\cdots,\gamma_T^k(\mathbf{x})\right)2^k. \tag{7}$$

The guarantees we obtain through the median estimator are formalized in the theorem below:

**Theorem 3.2.** *For the median estimator, there exists a $k^\star > 0$ and positive constant $\alpha$ such that for any $\Delta > 0$, if $T \geq \frac{1}{\alpha}\left(\log(2n/\Delta)\right)$ then with probability at least $(1-2\Delta)$,*

$$4p_\theta(\mathbf{x}) \geq \mathcal{L}_{Md}^{k^\star,T}(\mathbf{x}) \geq \frac{p_\theta(\mathbf{x})}{32(n+1)}$$

*Proof sketch:* The upper bound follows from the application of Markov's inequality to the positive random variable $\sum_{\mathbf{z}:A_t^k\mathbf{z}=b_t^k \mod 2} p_\theta(\mathbf{x},\mathbf{z})$ (first moments are bounded from Lemma 3.1) and $\gamma_t^k(\mathbf{x})$ lower bounds this sum. The lower bound of the above theorem extends a result from Theorem 2 of [15]. Please refer to the supplementary material for a detailed proof.

Hence, the rescaled variational solutions aggregated through a mean or median can provide tight bounds on the log-likelihood estimate for the observed data with high probability unlike the ELBO estimates in Eq. (1) and Eq. (2), which could be arbitrarily far from the true log-likelihood.

## 4 Algorithmic framework

In recent years, there have been several algorithmic advancements in variational inference and learning using *black-box* techniques [23]. These techniques involve a range of ideas such as the use of mini-batches, amortized inference, Monte Carlo gradient computation, etc., for scaling variational techniques to large data sets. See Section 6 for a discussion. In this section, we integrate random projections into a black-box algorithm for belief networks, a class of directed, discrete latent variable models. These models are especially hard to learn, since the "reparametrization trick" [17] is not applicable to discrete latent variables leading to gradient updates with high variance.

### 4.1 Model specification

We will describe our algorithm using the architecture of a sigmoid belief network (SBN), a multi-layer perceptron which is the basic building block for directed deep generative models with discrete latent variables [21]. A sigmoid belief network consists of $L$ densely connected layers of binary hidden units ($\mathbf{Z}^{1:L}$) with the bottom layer connected to a single layer of binary visible units ($\mathbf{X}$). The nodes and edges in the network are associated with biases and weights respectively. The state of the units in the top layer ($\mathbf{Z}^L$) is a sigmoid function ($\sigma(\cdot)$) of the corresponding biases. For all other layers, the conditional distribution of any unit given its parents is represented compactly by a non-linear activation of the linear combination of the weights of parent units with their binary state and an additive bias term. The generative process can be summarized as follows:

$$p(\mathbf{Z}_i^L = 1) = \sigma(b_i^L); \quad p(\mathbf{Z}_i^l = 1|\mathbf{z}^{l+1}) = \sigma(W^{l+1}\cdot\mathbf{z}^{l+1}+b_i^l); \quad p(\mathbf{X}_i = 1|\mathbf{z}^1) = \sigma(W^1\cdot\mathbf{z}^1+b_i^0)$$

In addition to the basic SBN design, we also consider the amortized inference setting. Here, we have an inference network with the same architecture as the SBN, but the feedforward loop running in the reverse direction from the input ($\mathbf{x}$) to the output $q(\mathbf{z}^L|\mathbf{x})$.

**Algorithm 1** VB-MCS: Learning belief networks with random projections.

---

**VB-MCS** (Mini-batches $\{\mathbf{x}^h\}_{h=1}^H$, Generative Network $(G, \theta)$, Inference Network $(I, \phi)$, Epochs $E$, Constraints $k$, Instances $T$)

   **for** $e = 1 : E$ **do**

      **for** $h = 1 : H$ **do**

         **for** $t = 1 : T$ **do**

            Sample $A \in \{0, 1\}^{k \times n} \overset{iid}{\sim} \text{Bernoulli}(\frac{1}{2})$ and $b \in \{0, 1\}^k \overset{iid}{\sim} \text{Bernoulli}(\frac{1}{2})$

            $C, b' \leftarrow \text{RowReduce}(A, b)$

            $\log \gamma_t^k(\mathbf{x}^h) \leftarrow \text{ComputeProjectedELBO}(\mathbf{x}^h, G, \theta, I, \phi, C, b')$

         $\log \mathcal{L}^{k,T}(\mathbf{x}^h) \leftarrow \log [\text{Aggregate}(\gamma_1^k(\mathbf{x}^h), \cdots, \gamma_T^k(\mathbf{x}^h))]$

         Update $\theta, \phi \leftarrow \text{StochasticGradientDescent}(-\log \mathcal{L}^{k,T}(\mathbf{x}^h))$

   **return** $\theta, \phi$

---

## 4.2 Algorithm

The basic algorithm for learning belief networks with augmented inference networks is inspired by the wake-sleep algorithm [13]. One key difference from the wake-sleep algorithm is that there is a single objective being optimized. This is typically the ELBO (see Eq. ( 1)) and optimization is done using stochastic mini-batch descent jointly over the model and inference parameters.

Training consists of two alternating phases for every mini-batch of points. The first step makes a forward pass through the inference network producing one or more samples from the top layer of the inference network, and finally, these samples complete a forward pass through the generative network. The reverse pass computes the gradient of the model and variational parameters with respect to the ELBO in Eq. (1) and uses these gradient updates to perform a gradient descent step on the ELBO.

We now introduce a black-box technique within this general learning framework, which we refer to as Variational Bayes on Monte Carlo Steroids (VB-MCS) due to the exponential sampling property. VB-MCS requires as input a data-dependent parameter $k$, which is the number of variables to constrain. At every training epoch, we first sample entries of a full-rank constraint matrix $A \in \{0, 1\}^{k \times n}$ and vector $b \in \{0, 1\}^k$ and then optimize for the objective corresponding to a projected variational inference problem defined in Eq. (5). This procedure is repeated for $T$ problem instances, and the individual likelihood estimates are aggregated using the mean or median based estimators defined in Eq. (6) and Eq. (7). The pseudocode is given in Algorithm 1.

For computing the projected ELBO, the inference network considers the marginal distribution of only $n - k$ free latent variables. We consider the mean-field family of approximations where the free latent variables are sampled independently from their corresponding marginal distributions. The remaining $k$ latent variables are specified by parity constraints. Using Gaussian elimination, the original linear system $A\mathbf{z} = b \mod 2$ is reduced into a row echleon representation of the form $C\mathbf{z} = b'$ where $C = [I_{kxk}|A']$ such that $A' \in \{0, 1\}^{k \times (n-k)}$ and $b' \in \{0, 1\}^k$. Finally, we read off the constrained variables as $z_j = \bigoplus_{i=k+1}^n c_{ji} z_i \oplus b'_j$ for $j = 1, 2, \cdots, k$ where $\oplus$ is the XOR operator.

## 5 Experimental evaluation

We evaluated the performance of VB-MCS as a black-box technique for learning discrete, directed latent variable models for images and documents. Our test-architecture is a simple sigmoid belief network with a single hidden layer consisting of 200 units and a visible layer. Through our experiments, we wish to demonstrate that the theoretical advantage offered by random projections easily translates into practice using an associated algorithm such as VB-MCS. We will compare a baseline sigmoid belief network (Base-SBN) learned using Variational Bayes and evaluate it against a similar network with parity constraints imposed on $k$ latent variables (henceforth, referred as $k$-SBN) and learned using VB-MCS. We now discuss some parameter settings below, which have been fixed with respect to the best validation performance of Base-SBN on the Caltech 101 Silhouettes dataset.

**Implementation details:** The prior probabilities for the latent layer are specified using autoregressive connections [10]. The learning rate was fixed based on validation performance to $3 \times 10^{-4}$ for the generator network and reduced by a factor of 5 for the inference network. Mini-batch size was fixed

Table 1: Test performance evaluation of `VB-MCS`. Random projections lead to improvements in terms of estimated negative log-likelihood and log-perplexity.

| Dataset | Evaluation Metric | Base | k=5 | k=10 | k=20 |
|---|---|---|---|---|---|
| **Vision:** Caltech 101 Silhouettes | NLL | 251.04 | **245.60** | 248.79 | 256.60 |
| **Language:** NIPS Proceedings | log-perplexity | 5009.79 | 4919.35 | **4919.22** | 4920.71 |

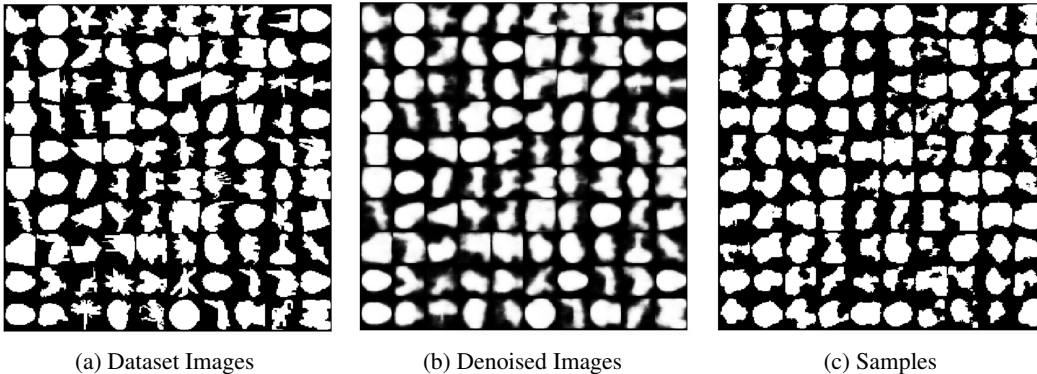

| (a) Dataset Images | (b) Denoised Images | (c) Samples |
|---|---|---|

Figure 1: Denoised images (center) of the actual ones (left) and sample images (right) generated from the best $k$-SBN model trained on the Caltech 101 Silhouettes dataset.

to 20. Regularization was imposed by early stopping of training after 50 epochs. The optimizer used is Adam [16]. For $k$-SBN, we show results for three values of $k$: 5, 10, and 20, and the aggregation is done using the median estimator with $T = 3$.

## 5.1 Generative modeling of images in the Caltech 101 Silhouettes dataset

We trained a generative model for silhouette images of $28 \times 28$ dimensions from the Caltech 101 Silhouettes dataset [3]. The dataset consists of 4,100 train images, 2,264 validation images and 2,307 test images. This is a particularly hard dataset due to the asymmetry in silhouettes compared to other commonly used structured datasets. As we can see in Table 1, the $k$-SBNs trained using `VB-MCS` can outperform the Base-SBN by several nats in terms of the negative log-likelihood estimates on the test set. The performance for $k$-SBNs dips as we increase $k$, which is related to the empirical quality of the approximation our algorithm makes for different $k$ values.

The qualitative evaluation results of SBNs trained using `VB-MCS` and additional control variates [19] on denoising and sampling are shown in Fig. 1. While the qualitative evaluation is subjective, the denoised images seem to smooth out the edges in the actual images. The samples generated from the model largely retain essential qualities such as silhouette connectivity and varying edge patterns.

## 5.2 Generative modeling of documents in the NIPS Proceedings dataset

We performed the second set of experiments on the latest version of the NIPS Proceedings dataset[4] which consists of the distribution of words in all papers that appeared in NIPS from 1988-2003. We performed a 80/10/10 split of the dataset into 1,986 train, 249 validation, and 248 test documents. The relevant metric here is the average perplexity per word for $D$ documents, given by $\mathcal{P} = \exp\left(\frac{-1}{D}\sum_{i=1}^{D}\frac{1}{L_i}\log p(\mathbf{x_i})\right)$ where $L_i$ is the length of document $i$. We feed in raw word counts per document as input to the inference network and consequently, the visible units in the generative network correspond to the (unnormalized) probability distribution of words in the document.

Table 1 shows the log-perplexity scores (in nats) on the test set. From the results, we again observe the superior performance of all $k$-SBNs over the Base-SBN. The different $k$-SBNs have comparable performance, although we do not expect this observation to hold true more generally for other

datasets. For a qualitative evaluation, we sample the relative word frequencies in a document and then generate the top-50 words appearing in a document. One such sampling is shown in Figure 2. The bag-of-words appears to be semantically reflective of coappearing words in a NIPS paper.

## 6 Discussion and related work

performance directional generated
concluding theoretically   program
     canada      sequences favors   feedforward
medicine    conversational      represents
engineering allowing internal happen fea finer
     aggregates     ctive parity press   discrete
graph compositionality      centers
     avenues markets corner      lil electronic
     model   smoothness minor jolla
     bottom newton coincidences   pertur
movements associations properties
     reversed consistent mussa likelihoods
     expands representing prototypical

Figure 2: Bag-of-words for a 50 word document sampled from the best $k$-SBN model trained on the NIPS Proceedings dataset.

There have been several recent advances in approximate inference and learning techniques from both a theoretical and empirical perspective. On the empirical side, the various black-box techniques [23] such as mini-batch updates [14], amortized inference [9] etc. are key to scaling and generalizing variational inference to a wide range of settings. Additionally, advancements in representational learning have made it possible to specify and learn highly expressive directed latent variable models based on neural networks, for e.g., [4, 10, 17, 19, 20, 24]. Rather than taking a purely variational or sampling-based approach, these techniques stand out in combining the computational efficiency of variational techniques with the generalizability of Monte Carlo methods [25, 26].

On the theoretical end, there is a rich body of recent work in hash-based inference applied to sampling [11], variational inference [15], and hybrid inference techniques at the intersection of the two paradigms [28]. The techniques based on random projections have not only lead to better algorithms but more importantly, they come with strong theoretical guarantees [5, 6, 7].

In this work, we attempt to bridge the gap between theory and practice by employing hash-based inference techniques to the learning of latent variable models. We introduced a novel bound on the marginal log-likelihood of directed latent variable models with discrete latent units. Our analysis extends the theory of random projections for inference previously done in the context of discrete, fully-observed log-linear undirected models to the general setting of both learning and inference in directed latent variable models with discrete latent units while the observed data can be discrete or continuous. Our approach combines a traditional variational approximation with random projections to get provable accuracy guarantees and can be used to improve the quality of traditional ELBOs such as the ones obtained using a mean-field approximation.

The power of black-box techniques lies in their wide applicability, and in the second half of the paper, we close the loop by developing VB-MCS, an algorithm that incorporates the theoretical underpinnings of random projections into belief networks that have shown tremendous promise for generative modeling. We demonstrate an application of this idea to sigmoid belief networks, which can also be interpreted as probabilistic autoencoders. VB-MCS simultaneously learns the parameters of the (generative) model and the variational parameters (subject to random projections) used to approximate the intractable posterior. Our approach can still leverage backpropagation to efficiently compute gradients of the relevant quantities. The resulting algorithm is scalable and the use of random projections significantly improves the quality of the results on benchmark data sets in both vision and language domains.

Future work will involve devising random projection schemes for latent variable models with continuous latent units and other variational families beyond mean-field [24]. On the empirical side, it would be interesting to investigate potential performance gains by employing complementary heuristics such as variance reduction [19] and data augmentation [8] in conjunction with random projections.

## Acknowledgments

This work was supported by grants from the NSF (grant 1649208) and Future of Life Institute (grant 2016-158687).

## Footnotes

[1] For brevity, we use binary random variables, although our analysis extends to discrete random variables.

[2] This is the typical case: randomly generated constraints can be linearly dependent, leading to larger buckets.

[3]Available at `https://people.cs.umass.edu/~marlin/data.shtml`

[4]Available at `http://ai.stanford.edu/~gal/`

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
