[Supplementary Material]

# A  Appendix

For notational convenience, let

$$Y_t^k(\mathbf{x}) = \sum_{\mathbf{z}: A_t^k \mathbf{z} = b_t^k \mod 2} p_\theta(\mathbf{x}, \mathbf{z})$$

denote a random variable representing the projected marginal likelihood, where the randomness is over the choice of the matrices $A_t^k, b_t^k$, and

$$\delta_t^{k, \mathcal{Q}}(\mathbf{x}) = \min_{q \in \mathcal{Q}} D_{KL}\big(q_\phi(\mathbf{z}|\mathbf{x}) || R_{A_t^k, b_t^k}^k [p_\theta(\mathbf{z}|\mathbf{x})]\big).$$

denote the minimum KL-divergence within an approximating family of distributions $\mathcal{Q}$ and the true posterior projected using $A_t^k, b_t^k$.

Before proving Theorem 3.1 and Theorem 3.2, we first extend an important result from earlier work to our setting.

## A.1  Extension of Theorem 2 from [Hsu et al., 2016]

**Lemma A.1.** *For any $\Delta > 0$, let $T \geq \frac{1}{\alpha}\left(\log(2n/\Delta)\right)$. Let $A_t^k \in \{0,1\}^{k \times n} \overset{iid}{\sim}$ Bernoulli$(\frac{1}{2})$ and $b_t^k \in \{0,1\}^k \overset{iid}{\sim}$ Bernoulli$(\frac{1}{2})$ for $k \in \{0,1,\ldots,n\}$ and $t \in \{1,\ldots,T\}$. Let $\mathcal{D}$ denote the set of degenerate (deterministic) probability distributions. Then there exists a positive constant $\alpha$ such that with probability at least $(1 - \Delta)$*

$$p_\theta(\mathbf{x})/32 \leq \sum_{k=0}^n \exp\left(\underset{t \in [T]}{Median}\left(-\delta_t^{k, \mathcal{D}}(\mathbf{x}) + \log Y_t^k(\mathbf{x})\right)\right) 2^{k-1} \leq 32 p_\theta(\mathbf{x}) \tag{8}$$

*i.e., it is a 32-approximation to $p_\theta(\mathbf{x})$.*

*Proof.* By definition,

$$\delta_t^{k, \mathcal{D}}(\mathbf{x}) = \min_{q \in \mathcal{D}} \sum_{\mathbf{z}: A_t^k \mathbf{z} = b_t^k \mod 2} q_\phi(\mathbf{z}|\mathbf{x})\big[\log q_\phi(\mathbf{z}|\mathbf{x}) - \log p_\theta(\mathbf{x}, \mathbf{z})\big] + \log Y_t^k(\mathbf{x}).$$

For a degenerate distribution, $q \in \mathcal{D}$, the entropy is zero and all its mass is at a single point. Hence,

$$\delta_t^{k, \mathcal{D}}(\mathbf{x}) = -\max_{q \in \mathcal{D}} \sum_{\mathbf{z}: A_t^k \mathbf{z} = b_t^k \mod 2} q_\phi(\mathbf{z}|\mathbf{x}) \cdot \log p_\theta(\mathbf{x}, \mathbf{z}) + \log Y_t^k(\mathbf{x})$$

$$= -1 \cdot \max_{\mathbf{z}: A_t^k \mathbf{z} = b_t^k \mod 2} \log p_\theta(\mathbf{x}, \mathbf{z}) + \log Y_t^k(\mathbf{x}).$$

Rearranging terms,

$$-\delta_t^{k, \mathcal{D}}(\mathbf{x}) + \log Y_t^k(\mathbf{x}) = \max_{\mathbf{z}: A_t^k \mathbf{z} = b_t^k \mod 2} \log p_\theta(\mathbf{x}, \mathbf{z}).$$

Substituting the above expression into Eq. (8), we get

$$\sum_{k=0}^n \exp\left(Median\left(\max_{\mathbf{z}: A_1^k \mathbf{z} = b_1^k \mod 2} \log p_\theta(\mathbf{x}, \mathbf{z}), \cdots, \max_{\mathbf{z}: A_T^k \mathbf{z} = b_T^k \mod 2} \log p_\theta(\mathbf{x}, \mathbf{z})\right)\right) 2^{k-1}$$

$$= \sum_{k=0}^n Median\left(\exp\left(\max_{\mathbf{z}: A_1^k \mathbf{z} = b_1^k \mod 2} \log p_\theta(\mathbf{x}, \mathbf{z})\right), \cdots, \exp\left(\max_{\mathbf{z}: A_T^k \mathbf{z} = b_T^k \mod 2} \log p_\theta(\mathbf{x}, \mathbf{z})\right)\right) 2^{k-1}$$

$$= \sum_{k=0}^n Median\left(\max_{\mathbf{z}: A_1^k \mathbf{z} = b_1^k \mod 2} p_\theta(\mathbf{x}, \mathbf{z}), \cdots, \max_{\mathbf{z}: A_T^k \mathbf{z} = b_T^k \mod 2} p_\theta(\mathbf{x}, \mathbf{z})\right) 2^{k-1}.$$

The result then follows directly from Theorem 1 from [Ermon et al., 2013b]. $\qquad\square$

## A.2 Proof of Theorem 3.1: Upper bound based on mean aggregation

From the non-negativity of KL divergence we have that for any $q \in Q$,

$$\log Y_t^k(\mathbf{x}) \geq \sum_{\mathbf{z}:A_t^k \mathbf{z}=b_t^k \mod 2} q_\phi(\mathbf{z}|\mathbf{x})\big[\log p_\theta(\mathbf{x},\mathbf{z}) - \log q_\phi(\mathbf{z})\big]$$

$$\geq \max_{q \in Q}\left(\sum_{\mathbf{z}:A_t^k \mathbf{z}=b_t^k \mod 2} q_\phi(\mathbf{z}|\mathbf{x})\big[\log p_\theta(\mathbf{x},\mathbf{z}) - \log q_\phi(\mathbf{z})\big]\right)$$

Exponentiating both sides,

$$Y_t^k(\mathbf{x}) \geq \exp\left(\max_{q \in Q}\left(\sum_{\mathbf{z}:A_t^k \mathbf{z}=b_t^k \mod 2} q_\phi(\mathbf{z}|\mathbf{x})\big[\log p_\theta(\mathbf{x},\mathbf{z}) - \log q_\phi(\mathbf{z})\big]\right)\right) \overset{\text{def}}{=} \gamma_t^k(\mathbf{x}). \quad (9)$$

Taking an expectation on both sides w.r.t $A_t^k, b_t^k$,

$$\mathbb{E}_{A_t^k, b_t^k}[Y_t^k(\mathbf{x})] \geq \mathbb{E}_{A_t^k, b_t^k}[\gamma_t^k(\mathbf{x})]$$

Using Lemma 3.1, we get:

$$\mathbb{E}_{A_t^k, b_t^k}[\gamma_t^k(\mathbf{x})] \leq 2^{-k} p_\theta(\mathbf{x})$$

## A.3 Proof of Theorem 3.2: Upper bound based on median aggregation

From Markov's inequality, since $Y_t^k(\mathbf{x})$ is non-negative,

$$\mathbb{P}\left[Y_t^k(\mathbf{x}) \geq c\mathbb{E}[Y_t^k(\mathbf{x})]]\right] \leq \frac{1}{c}.$$

Using Lemma 3.1,

$$\mathbb{P}\left[Y_t^k(\mathbf{x})2^k \geq cp_\theta(\mathbf{x})\right] \leq \frac{1}{c}.$$

Since $Y_t^k(\mathbf{x}) \geq \gamma_t^k(\mathbf{x})$ from Eq. (9), setting $c = 4$ and $k = k^\star$ we get

$$\mathbb{P}\left[\gamma_t^{k^\star}(\mathbf{x})2^{k^\star} \geq 4p_\theta(\mathbf{x})\right] \leq \frac{1}{4}. \quad (10)$$

From Chernoff's inequality, if for any non-negative $\epsilon \leq 0.5$,

$$\mathbb{P}\left[\gamma_t^{k^\star}(\mathbf{x})2^{k^\star} \geq 4p_\theta(\mathbf{x})\right] \leq \left(\frac{1}{2} - \epsilon\right) \quad (11)$$

then,

$$\mathbb{P}\left[4p_\theta(\mathbf{x}) \leq Median\left(\gamma_1^{k^\star}(\mathbf{x}), \cdots, \gamma_T^{k^\star}(\mathbf{x})\right)2^{k^\star}\right] \leq \exp(-2\epsilon^2 T) \quad (12)$$

From Eq. (10) and Eq. (11), $\epsilon \leq 0.25$. Hence, taking the complement of Eq. (12) and given a positive constant $\alpha \leq 0.125$ such that for any $\Delta > 0$, if $T \geq \frac{1}{\alpha}\log(2n/\Delta) \geq \frac{1}{\alpha}\log(1/\Delta)$, then

$$\mathbb{P}\left[4p_\theta(\mathbf{x}) \geq Median\left(\gamma_1^{k^\star}(\mathbf{x}), \cdots, \gamma_T^{k^\star}(\mathbf{x})\right)2^{k^\star}\right] \geq 1 - \Delta.$$

## A.4 Proof of Theorem 3.2: Lower bound based on median aggregation

Since the conditions of Lemma A.1 are satisfied, we know that Eq. (8) holds with probability at least $1 - \delta$. Also, since the terms in the sum are non-negative we have that the maximum element is at least $1/(n+1)$ of the sum. Hence,

$$\max_k \exp\left(Median\left(-\delta_1^{k,\mathcal{D}}(\mathbf{x}) + \log Y_1^k(\mathbf{x}), \cdots, -\delta_T^{k,\mathcal{D}}(\mathbf{x}) + \log Y_T^k(\mathbf{x})\right)\right)2^{k-1} \geq \frac{1}{32}p_\theta(\mathbf{x})\frac{1}{n+1}. \quad (13)$$

Therefore, there exists $k^\star$ (corresponding to the $\arg\max$ in Eq. (13)) such that

$$Median\left(-\delta_1^{k^\star,\mathcal{D}}(\mathbf{x}) + \log Y_1^{k^\star}(\mathbf{x}), \cdots, -\delta_T^{k^\star,\mathcal{D}}(\mathbf{x}) + \log Y_T^{k^\star}(\mathbf{x})\right) + (k^\star - 1)\log 2 \geq -\log 32 + \log p_\theta(\mathbf{x}) - \log(n+1).$$

Since $\mathcal{D} \subseteq \mathcal{Q}$, we also have

$$\delta_t^{k^\star,\mathcal{Q}}(\mathbf{x}) \leq \delta_t^{k^\star,\mathcal{D}}(\mathbf{x}).$$

Thus,

$$Median\left(-\delta_1^{k^\star,\mathcal{Q}}(\mathbf{x}) + \log Y_1^{k^\star}(\mathbf{x}), \cdots, -\delta_T^{k^\star,\mathcal{Q}}(\mathbf{x}) + \log Y_T^{k^\star}(\mathbf{x})\right) + (k^\star - 1)\log 2 \geq -\log 32 + \log p_\theta(\mathbf{x}) - \log(n+1).$$

From Eq. (5), note that

$$\log \gamma_t^k(\mathbf{x}) = \max_{q \in \mathcal{Q}} \sum_{\mathbf{z}:A_t^k\mathbf{z}=b_t^k \bmod 2} q_\phi(\mathbf{z}|\mathbf{x})\left(\log p_\theta(\mathbf{x},\mathbf{z}) - \log q_\phi(\mathbf{z}|\mathbf{x})\right)$$

$$= \max_{q \in \mathcal{Q}} \sum_{\mathbf{z}:A_t^k\mathbf{z}=b_t^k \bmod 2} q_\phi(\mathbf{z}|\mathbf{x})\left(\log p_\theta(\mathbf{z}|\mathbf{x}) - \log q_\phi(\mathbf{z}|\mathbf{x})\right) + \log p_\theta(\mathbf{x})$$

$$= -\delta_t^{k,\mathcal{Q}}(\mathbf{x}) + \log Y_t^k(\mathbf{x}).$$

Plugging in we get,

$$Median\left(\log \gamma_1^{k^\star}(\mathbf{x}), \cdots, \log \gamma_T^{k^\star}(\mathbf{x})\right) + (k^\star - 1)\log 2 \geq -\log 32 - \log(n+1) + \log p_\theta(\mathbf{x})$$

and also

$$Median\left(\log \gamma_1^{k^\star}(\mathbf{x}), \cdots, \log \gamma_T^{k^\star}(\mathbf{x})\right) + k^\star \log 2 \geq -\log 32 - \log(n+1) + \log p_\theta(\mathbf{x}).$$

with probability at least $1 - \Delta$.

$$Median\left(\gamma_1^{k^\star}(\mathbf{x}), \cdots, \gamma_T^{k^\star}(\mathbf{x})\right) 2^{k^\star} \geq \frac{p_\theta(\mathbf{x})}{32(n+1)}.$$

Combining the lower and upper bounds, we get

$$4p_\theta(\mathbf{x}) \geq \mathcal{L}_{Md}^{k^\star,T}(\mathbf{x}) \geq \frac{p_\theta(\mathbf{x})}{32(n+1)}$$

with probability at least $1 - 2\Delta$ by union bound by choosing a small enough value for $\alpha$.

## A.5 Proof of Theorem 3.1: Lower bound based on mean aggregation

We first prove a useful inequality relating to the mean and median of non-negative reals.

**Lemma A.2.** *For a set of non-negative reals* $F = \{f_i\}_{i=1}^\ell$,

$$\frac{1}{\ell}\sum_{i=1}^\ell f_i \geq \frac{1}{2}Median(F).$$

*Proof.* Without loss of generality, we assume for notational convenience that elements in $F$ are sorted by their indices, i.e., $f_1 \leq f_2 \cdots \leq f_\ell$. By definition of median, we have for all $i \in \{\lfloor \ell/2 \rfloor, \lfloor \ell/2 \rfloor + 1, \ldots, \ell\}$

$$f_i \geq Median(F).$$

Adding all the above inequalities, we get

$$\sum_{i=\lfloor \ell/2 \rfloor}^\ell f_i \geq \left\lceil \frac{\ell}{2} \right\rceil Median(F).$$

Since all $f_i$ are non-negative,

$$\sum_{i=1}^\ell f_i \geq \left\lceil \frac{\ell}{2} \right\rceil Median(F).$$

The median of non-negative reals is also non-negative, and hence,

$$\sum_{i=1}^\ell f_i \geq \frac{\ell}{2}Median(F)$$

finishing the proof. □

Substituting for $F$ in the above lemma with $\{\gamma_t^{k^\star}(\mathbf{x})\}_{t=1}^T$, we get

$$\frac{1}{T}\sum_{t=1}^T \gamma_t^{k^\star}(\mathbf{x}) \geq \frac{1}{2}Median\left(\gamma_1^{k^\star}(\mathbf{x}), \cdots, \gamma_T^{k^\star}(\mathbf{x})\right).$$

Now using the lower bound in Theorem 3.2, with probability at least $1 - 2\Delta$,

$$\frac{1}{T}\sum_{t=1}^T \gamma_t^{k^\star}(\mathbf{x}) \cdot 2^{k^\star} \geq \frac{p_\theta(\mathbf{x})}{64(n+1)}.$$