[Reviews · NeurIPS 2016]

Reviewer 1

Summary

The authors describe an application of recent projection-based stochastic bounds on the partition function to improve the lower bound on the marginal likelihood used for training a latent variable model, such as a deep generative model. Pros: elegant and potentially important application of recent theory to hot problems in learning deep latent variable models; promising experimental results Cons: paper detail is spent on minor adaptation of recent work, and too much detail is omitted from the practical application of those ideas to the learning process

Qualitative Assessment

The concept seems solid and the authors' execution seems promising. It is intuitive that an improved estimate of the likelihood should also improve training. My main concerns are: (1) The theory developed appears to rest very strongly on [13]; while the authors say they "extend" [13] to bounding the marginal likelihood of directed latent variable models, this may not itself be a significantly novel contribution. (2) The application of the main idea to black-box learning and test likelihood for deep generative models do not seem sufficiently well explained; it is mainly a few paragraphs on page 6. The pseudocode in the supplement helps, but should not be required to understand the paper. Similarly, the discussion is often vague (for example Section 4.2.1). I feel that Section 4's practical application of the theory is the main contribution of the work; if the authors agree, it needs a more detailed treatment and discussion in the draft.

Confidence in this Review

1-Less confident (might not have understood significant parts)


Reviewer 2

Summary

This paper proposes an extension of recently proposed random-projection methods to models with latent variables using variational inference. The authors are able to derive novel estimates of marginal likelihoods, and to bound the error of those estimates.

Qualitative Assessment

I found the approach and the paper quite interesting—it's a very different way of doing approximate posterior inference, and it's great that some kind of theoretical analysis is possible. I'm not an expert on these new discrete random projection results, but this seems like a nice application of them. I was less impressed with the empirical study. For one thing, it's not at all clear that there's enough information to reproduce these results. "Many control variate techniques were employed" is not sufficient detail. The quantitative results are suggestive, but not overwhelmingly compelling, and the qualitative results are hard to interpret. A few more specific comments: You should consider changing the title/algorithm name. The meaning of the colloquial phrase "x on steroids" as "a stronger version of x" is clear enough, but an academic paper title isn't the most appropriate place to make light of substance abuse. Also, steroid abuse comes with some nasty side effects that you probably don't mean to associate with your algorithm. The bounds in section 3.2 feel very loose at first glance---their relative tightness might be clearer if they were expressed in terms of log p_theta as one traditionally does in variational inference derivations. Being off by a factor of 3200 in a 100-latent-variable model sounds pretty bad, even if that's only 8 nats. Line 83: I wonder if this statement conflates the ability of deep networks to produce multimodal marginal distributions p(x) rather than multimodal conditional distributions p(z|x). Line 95: [0,n] should be {0,...,n}, since k is integer.

Confidence in this Review

2-Confident (read it all; understood it all reasonably well)


Reviewer 3

Summary

The authors describe a method for variational inference in latent variable models that is motivated by importance sampling and recent work on random perturbations and hashing. Additionally, the authors provide bounds (that depend only on the number of latent variables) on their estimator that hold with high probability, in contrast to typical variational methods which can be arbitrarily bad. Finally, the authors describe how to apply these theoretical tools to the design of variational learning algorithms in a variety of settings.

Qualitative Assessment

I liked the paper and the results, but I do think that some of the high level proofs could be improved. I would also have liked the authors to comment a bit more about the hardness of mean field and how that affects practical performance. General comments/typos: -line 19, "Variational approximations are particularly suitable for directed models" -> more so than undirected models? -line 70, "variational inference is optimizes" -> "variational inference optimizes" -line 133, "i.e. distributions" -> "i.e., distributions" -line 134, is this the same Q as on line 130? -line 290, "for e.g.," -> "e.g.,"

Confidence in this Review

2-Confident (read it all; understood it all reasonably well)


Reviewer 4

Summary

The authors suggest a new method to perform approximate inference in latent discrete variable models, by adding random projections (random parity check constraints) to the posterior and correcting for the bias introduced by the subsampling. They prove that if one can solve the randomly-projected problem, one can recover the true posterior with high accuracy.

Qualitative Assessment

An interesting technical paper. More space should have been devoted to (ideally geometric) intuition behind the theory, and strengthening the experimental section, which is on the weaker side right now (few experiments, weak baselines). - The paper follows [13] quite closely - this is not necessarily a deal breaker, but it would benefit from clearly highlighting the contribution from previous similar work. - The paper main idea is that to divide the main problem into subproblems, for which we have removed many points from the posterior (by adding random parity check constraints) and appropriately scaled the estimates to compensate for those missing points. The implication would be that if we can solve the subproblems, then we have an excellent approximation to the log-partition function of the original problem. But it is not clear to me the subproblems are in fact significantly easier to solve. One intuition would be that by removing points at random, each subproblem can be slightly less complex (less multi-modal?) and the posterior can focus on those fewer modes instead (this seems suggested in lines 31-33, 85-88). I could very well be wrong, but it's not clear to me this should be occurring - the parity check constraints are not going to remove a particular part of space, but will appear to remove points from the posterior quite uniformly. In other words, instead of removing a bunch of modes, it seems to me it should make 'swiss cheese' from the posterior, which might not be significantly easier to approximate as a result.

Confidence in this Review

2-Confident (read it all; understood it all reasonably well)


Reviewer 5

Summary

This work extends hash-based inferences techniques to the learning of latent models. It bridges the gap between existent the theory for hash-based inference for discrete latent variable models and the existent practical inference techniques (sampling and variational methods). The authors provide a theoretical bound on the marginal likelihood of the latent variable models when using hash-based inference then apply it to learning sigmoid belief networks where they claim an improvement as compared to existing benchmarks.

Qualitative Assessment

This paper provides theoretical bounds that are tighter than existing variational bounds for the problem of learning latent variable models. The authors extend applied existing theory of hash-based learning and amortized inference to design a black-box learning algorithm. They later applied it to learning a Sigmoid Belief Network. The main advantage to this approach seems to be the partitioning of the search space for posterior distributions into buckets/subsets that are faster to search than with a typical sampling method. The proposed inference scheme then leverages mean-field inference (used heavily in the context of variational inference) within each subset. One of the main technical contributions is the tighter bound on the likelihood using two aggregate estimators which was an extension of an existing work (specific to undirected graphical models) to the directed models setting. However, it was not intuitive how this was more advantageous to stochastic (called mini-batch in the paper) or batch variational inference schemes. While the general variational bounds on the marginal likelihood are provenly tighter, we would have appreciated a less terse explanation (in comparison to mini-batch + mean-field). Additionally, the empirical analysis seemed lacking as there weren't enough details regarding recent developments and heuristics in the domain were discarded as complementary rather than competing. However, it is clear that the goal of the paper was to prove the superiority of this novel hash-based inference scheme to the common sampling or variational (they used stochastic variational inference for learning the ELBO of the basic SBN unless I misunderstood?) and for that their analysis might be adequate. Overall, despite a few typographical errors, this paper was written eloquently and the methodology was presented adequately and the analysis of the results was quite appropriate.

Confidence in this Review

2-Confident (read it all; understood it all reasonably well)


Reviewer 6

Summary

The paper presents a hybrid variational/monte carlo inference technique for improving variational lower bound. The method has theoretical guarantees for tightness. Experimental evaluations on neural nets (sigmoid belief network) is presented for image and text modeling.

Qualitative Assessment

Several things to note: 1. Given the projection step and multiple iterations for collecting lower bound values, how is the running time affected? 2. Some additional plots comparing lower bound values might be useful. In particular, given the numbers in your experimental section, it seems like the base variational inference bound is not that loose either. Some additional plots making the improvement clear will make the presentation better (maybe in the supplement?).

Confidence in this Review

2-Confident (read it all; understood it all reasonably well)